# Edoxaban Exerts Antioxidant Effects Through FXa Inhibition and Direct Radical-Scavenging Activity

**DOI:** 10.3390/ijms20174140

**Published:** 2019-08-24

**Authors:** Yuki Narita, Kana Hamamura, Mami Kashiyama, Sara Utsumi, Yutaka Kakizoe, Yuki Kondo, Yoichi Ishitsuka, Hirofumi Jono, Tetsumi Irie, Masashi Mukoyama, Hideyuki Saito, Daisuke Kadowaki, Sumio Hirata, Kenichiro Kitamura

**Affiliations:** 1Department of Pharmacy, Kumamoto University Hospital, 1-1-1 Honjo, Chuo-ku, Kumamoto 860-8556, Japan; 2Department of Clinical Pharmaceutical Sciences, Graduate School of Pharmaceutical Sciences, Kumamoto University, 1-1-1 Honjo, Chuo-ku, Kumamoto 860-8556, Japan; 3Department of Clinical Pharmacology, Graduate School of Pharmaceutical Sciences, Kumamoto University, 5-1 Oe-honmachi, Chuo-ku, Kumamoto 862-0973, Japan; 4Department of Nephrology, Graduate School of Medical Sciences, Kumamoto University, 1-1-1 Honjo, Chuo-ku, Kumamoto 860-8556, Japan; 5Department of Clinical Chemistry and Informatics, Graduate School of Pharmaceutical Sciences, Kumamoto University, 5-1 Oe-honmachi, Chuo-ku, Kumamoto 862-0973, Japan; 6Department of Clinical Pharmaceutics, Faculty of Pharmaceutical Sciences, Sojo University, 4-22-1 Ikeda, Nishi-ku, Kumamoto 860-0082, Japan; 7Department of Internal Medicine III, Faculty of Medicine, University of Yamanashi, 1110 Shimokato, Chuo-shi, Yamanashi 409-3898, Japan

**Keywords:** oxidative stress, reactive oxygen species, antioxidant effect, edoxaban, factor Xa, protease-activated receptor 2

## Abstract

The interplay between oxidative stress, inflammation, and tissue fibrosis leads to the progression of chronic kidney disease (CKD). Edoxaban, an activated blood coagulation factor Xa (FXa) inhibitor, ameliorates kidney disease by suppressing inflammation and tissue fibrosis in animal models. Interestingly, rivaroxaban, another FXa inhibitor, suppresses oxidative stress induced by FXa. Thus, FXa inhibitors could be multitargeted drugs for the three aforementioned risk factors for the progression of CKD. However, the exact mechanism responsible for eliciting the antioxidant effect of FXa inhibitors remains unclear. In this study, the antioxidant effect of edoxaban was evaluated. First, the intracellular antioxidant properties of edoxaban were evaluated using human proximal tubular cells (HK-2 cells). Next, direct radical scavenging activity was measured using the electron spin resonance and fluorescence analysis methods. Results show that edoxaban exhibited antioxidant effects on oxidative stress induced by FXa, indoxyl sulfate, and angiotensin II in HK-2 cells, as well as the FXa inhibitory activity, was involved in part of the antioxidant mechanism. Moreover, edoxaban exerted its antioxidative effect through its structure-specific direct radical scavenging activity. Edoxaban exerts antioxidant effects by inhibiting FXa and through direct radical-scavenging activity, and thus, may serve as multitargeted drugs for the three primary risk factors associated with progression of CKD.

## 1. Introduction

Chronic kidney disease (CKD) not only causes kidney damage but also causes high cardiovascular morbidity and mortality. The morbidity rate is estimated at 10.4% among men and 11.8% among women worldwide. Hence, CKD is an important global public health concern [1]. The interplay between oxidative stress, inflammation, and tissue fibrosis leads to the progression of CKD [2,3]. Therefore, multitarget therapies for these risk factors are preferred to suppress the progression of kidney disease.

Activated blood coagulation factor Xa (FXa) inhibitors have attracted attention as new oral anticoagulants for overcoming the disadvantages of warfarin, which has been used worldwide as the only oral anticoagulant. FXa not only participates in blood coagulation as a central factor in the blood coagulation cascade but also activates the protease-activated receptor (PAR) on the cell surface and causes inflammatory diseases. Four subtypes of PAR have been identified [4], among them, PAR1 and PAR2 are activated by FXa [5]. Notably, PAR2 has been reported to be closely associated with inflammation [6,7]. Interestingly, it has been reported that renal FX and PAR2 mRNA and urinary FXa activity are increased, although there is no change in the activity of plasma FXa in mouse models of diabetic nephropathy [8] and unilateral ureteral ligation renal interstitial fibrosis [9]. Moreover, edoxaban, an FXa inhibitor, reportedly ameliorates kidney disease by suppressing not only inflammation but also renal fibrosis in the two aforementioned mouse models. In addition, rivaroxaban, another FXa inhibitor, has been reported to suppress oxidative stress induced by FXa at the abdominal aortic aneurysm site and to suppress oxidative stress enhanced by citrated plasma and advanced glycation end-products [10,11]. Thus, FXa inhibitors could be multitargeted drugs for the three aforementioned risk factors for the development of kidney disease. However, the mechanisms of the antioxidant effect of FXa inhibitors remain unclear.

In this study, the antioxidant properties of edoxaban, an FXa inhibitor, against oxidative stress induced by various stimulants and its direct radical scavenging activity were evaluated.

## 2. Results

### 2.1. Effect of Edoxaban on Intracellular Reactive Oxygen Species Production Induced by FXa in HK-2 Cells

Since overproduction of reactive oxygen species (ROS) causes oxidative stress, the antioxidant effect of edoxaban against FXa-induced intracellular ROS production was assessed using human proximal tubular cells (HK-2 cells). Results show that FXa induced oxidative stress at 0.5 IU/mL, and edoxaban significantly inhibited FXa-induced intracellular ROS production under pathophysiological conditions in HK-2 cells (Figure 1). Consistently, the PAR-2 inhibitor Phe-Ser-Leu-Leu-Arg-Tyr-NH2 (FSLLRY-NH2) and nicotinamide adenine dinucleotide phosphate (NADPH) oxidase inhibitor diphenyleneiodonium chloride (DPI) significantly inhibited FXa-induced intracellular ROS production.

### 2.2. Effect of Edoxaban on Intracellular Reactive Oxygen Species Production Induced by Indoxyl Sulfate and Angiotensin II in HK-2 Cells

Previous studies have shown that edoxaban exhibited renoprotective effects in mouse models of diabetic nephropathy [8] and unilateral ureteral ligation renal interstitial fibrosis [9]. We, therefore, examined the antioxidant activity of edoxaban against intracellular ROS production in HK-2 cells induced by indoxyl sulfate (IS) and angiotensin II (AII), which stimulate oxidative stress at the time of renal injury. As shown in Figure 2, IS significantly increased intracellular ROS production, while edoxaban inhibited IS-induced intracellular ROS production in a dose-dependent manner (Figure 2a). Moreover, AII significantly increased intracellular ROS production, while 100 µM edoxaban significantly inhibited AII-induced intracellular ROS production (Figure 2b). DPI also significantly inhibited IS and AII-induced intracellular ROS production.

### 2.3. Radical Scavenging Activity of Edoxaban Against Hydroxyl Radical and Hydrogen Peroxide Decomposition Ability

The radical scavenging activity of edoxaban for the highly cytotoxic hydroxyl radical (•OH) generated by UV photolysis of hydrogen peroxide (H_2_O_2_) was evaluated by the electron spin resonance (ESR) method. The signal intensity was calculated as the ratio of 5,5-Dimethyl-1-pyrroline-*N*-oxide (DMPO)/•OH adduct and manganese marker and expressed as a ratio compared to the control. As shown in Figure 3a,b, edoxaban did not affect the ESR spectrum in the concentration range of 0.1–100 µ. Furthermore, the decomposition ability to H_2_O_2_, which generates •OH by UV photolysis and is itself a ROS, was evaluated. The results showed that edoxaban did not affect the decomposition of H_2_O_2_ (Figure 3c).

### 2.4. Radical Scavenging Activity of Edoxaban against Peroxynitrite

To evaluate the radical scavenging activity of edoxaban against peroxynitrite (ONOO^−^), which is as highly cytotoxic as •OH, the scavenging activity against ONOO^−^ generated by degradation of 3-(4-morpholinyl) sydnonimine hydrochloride (SIN-1) was evaluated by monitoring the oxidation of dihydrorhodamine123 (DHR123). As shown in Figure 4, edoxaban scavenged ONOO^−^ at concentrations of 10 to 100 µM.

### 2.5. Radical Scavenging Activity of Edoxaban against Superoxide Radical

To evaluate the radical scavenging activity of edoxaban against superoxide radical (O_2_˙^−^), which is a precursor of •OH and ONOO^−^, the scavenging activity against O_2_˙^−^ produced by the xanthine–xanthine oxidase system was evaluated by the ESR method. Edoxaban scavenged O_2_˙^−^ at a concentration of 100 µM and tended to exert its scavenging activity at concentrations of 0.1 to 10 µM (Figure 5a,b).

## 3. Discussion

In the present study, we demonstrated that edoxaban alleviated the oxidative stress induced by FXa and that FXa inhibitory activity was involved in part of the antioxidant mechanism. Furthermore, edoxaban showed antioxidative effect through structure-specific direct radical scavenging activity, which differed from the FXa inhibitory activity. Interestingly, we showed that edoxaban reduced oxidative stress induced by IS and AII.

Rivaroxaban has been reported to suppress oxidative stress induced by FXa [12]. FXa is known to activate PAR2 and cause inflammation, tissue fibrosis, and cell proliferation. Additionally, recent studies have shown that PAR2 signaling is involved in various diseases, such as cancer [13], arteriosclerosis [14,15], and fibrosis [16,17], and thus, we hypothesized that the FXa-PAR2 pathway is involved in oxidative stress. Therefore, we examined the effect of FSLLRY, a PAR2-selective inhibitor, to clarify the relationship between the FXa-PAR2 pathway and oxidative stress. We found that FSLLRY inhibited FXa-induced intracellular ROS production (Figure 1). These results suggest that the FXa-PAR2 pathway produces oxidative stress, and edoxaban acts as an antioxidant by blocking the pathway. Moreover, the antioxidative effect of FXa inhibitors, including edoxaban, is a class effect.

We next examined the antioxidant effect of edoxaban against IS and AII, which are associated with oxidative stress at the time of renal injury. Interestingly, edoxaban had an inhibitory effect on IS and AII-induced intracellular ROS production (Figure 2). IS activates NADPH oxidase and increases ROS, such as O_2_˙^−^, in many cell types, such as renal tubular epithelial cells [18,19] and vascular endothelial cells [20]. Similarly, AII is a representative activator of NADPH oxidase and is known to increase oxidative stress [21]. DPI, which selectively inhibits NADPH oxidase, significantly inhibited IS and AII-induced ROS production. We also predicted that edoxaban did not inhibit IS and AII-induced intracellular ROS production when FXa was not involved. Unexpectedly, edoxaban maintained its inhibitory activity against IS- and AII-induced intracellular ROS production. No FXa was present in the solution in IS- and AII-induced ROS production experiments, and no studies have demonstrated a relationship between IS or AII and FXa. Thus, edoxaban may have a pleiotropic effect differing from its inhibitory action.

Therefore, we predicted that the molecular structure was involved in the pleiotropic effect of edoxaban and evaluated the structure-specific direct radical scavenging activity. Among ROS, O_2_˙^−^ is less reactive than other radicals and does not substantially contribute to oxidative damage. However, because it is a highly cytotoxic precursor of H_2_O_2_ or •OH, O_2_˙^−^ is a major cause of oxidative stress. Moreover, O_2_˙^−^ mediates pro-oxidative and pro-inflammatory changes in endothelial cells. Therefore, O_2_˙^−^ is thought to impair vascular endothelial function and cause organ damage, such as cardiovascular disease, hypertension, and kidney disease [22,23]. Additionally, O_2_˙^−^ reacts with NO to form highly toxic ONOO^−^ [24]. ONOO^−^ causes cell damage and is reported to be involved in the onset of lifestyle-related diseases; along with •OH, ONOO^−^ causes oxidation and damage of biomolecules [25]. In the present study, edoxaban showed radical scavenging activity for ONOO^−^ (Figure 4) and O_2_˙^−^ (Figure 5), although radical scavenging activity for •OH and H_2_O_2_ was not observed (Figure 3). As described above, IS and AII activate NADPH oxidase to increase ROS, such as O_2_˙^−^. Therefore, edoxaban inhibited intracellular ROS production possibly by directly scavenging O_2_˙^−^_,_ and ONOO^−^ increased by IS and AII. Moreover, FXa-induced ROS production was inhibited by DPI. The pathway downstream of PAR2 has been reported to produce oxidative stress through NADPH oxidase [26], with O_2_˙^−^ predicted as the main ROS produced. Thus, the antioxidant activity of edoxaban in this experimental system may involve not only antioxidant activity by blocking the FXa-PAR2 pathway but also a structure-specific direct radical trapping activity.

Because FXa inhibitors, including edoxaban, are excreted from the kidneys, assessing the pharmacological changes that occur with decreased renal function is important [27]. The renal excretion rate of edoxaban is 50%, which increases its blood concentration in CKD patients, in turn, increasing the risk of bleeding. Therefore, according to the package insert, 60 mg once a day for creatinine clearance (Ccr) of 50 mL/min or more and 30 mg once a day for Ccr of 15–50 mL/min is recommended for CKD patients. Furthermore, patients with end-stage renal disease with Ccr of less than 15 mL/min are contraindicated because they have no experience in use and may pose a risk of bleeding, exceeding the benefits. Therefore, effective use of edoxaban, including the antioxidant effects revealed in this study, requires appropriate consideration of renal function.

The present study demonstrated that edoxaban mitigated the oxidative stress induced by FXa, IS, and AII. Moreover, its mechanism was exerted through the FXa inhibitory action of edoxaban as well as structure-specific radical trapping activity towards O_2_˙^−^ and ONOO^−^. Edoxaban exerts antioxidant effects through FXa inhibitory activity and direct radical-scavenging activity, and thus, may serve as a multitargeted drug against the three primary risk factors (oxidative stress, inflammation, and tissue fibrosis) associated with the development of CKD.

## 4. Materials and Methods

### 4.1. Materials

Edoxaban tosylate hydrate was a kind gift from Daiichisankyo Pharmaceutical, Co., Ltd. (Tokyo, Japan). HK-2 cells were purchased from the American Type Culture Collection (Manassas, VA, USA). Fetal bovine serum was purchased from Corning, Inc (Corning, NY, USA). Xanthine, xanthine oxidase, DHR123, IS, and DPI were purchased from Sigma-Aldrich (St. Louis, MO, USA). Luminol was purchased from Nacalai Tesque, Inc. (Kyoto, Japan). Ascorbic acid, AII, H_2_O_2_, diethylenetriamine-pentaacetic acid, and SIN-1 were purchased from Wako Pure Chemical Industries, Ltd. (Osaka, Japan). DMPO was purchased from Labotec, Co., Ltd. (Tokyo, Japan). 5-(and-6)-Chloromethyl-2′,7′-dichlorodihydrofluorescein diacetate (CM-H_2_DCFDA) was purchased from Invitrogen (Carlsbad, CA, USA). Factor Xa was purchased from Enzyme Research Laboratories (South Bend, IN, USA). FSLLRY-NH2 was purchased from Funakoshi, Co., Ltd. (Tokyo, Japan). All other chemicals were of the highest grade available from commercial sources.

### 4.2. Cell Culture

The HK-2 cells, which immortalized the proximal tubule epithelial cell line from normal adult human kidneys, were used [28]. The HK-2 cells were cultured in Dulbecco’s modified Eagle’s medium (D-MEM)/Ham’s F-12 medium containing 10% fetal bovine serum under standard cell culture conditions (humidified atmosphere, 5% CO_2_, 37 °C).

### 4.3. Measurement of Reactive Oxygen Species Production

To measure ROS production, CM-H_2_DCFDA, a ROS-sensitive fluorescent dye, was used as a probe. HK-2 cells were cultured in 96-well plates (5 × 10^3^ cells/well) at 37 °C for 24 h. The medium was changed to D-MEM/Ham’s F-12 medium containing FXa (0.5 IU/mL), IS (500 µM), or AII (25 µM) followed by incubation for 24 h. Dulbecco’s phosphate buffered saline (D-PBS) containing CM-H_2_DCFDA (5 µM) was added to each well, and then the cells were incubated for 30 min to incorporate CM-H_2_DCFDA into the cells. After removing the D-PBS from the wells, the HK-2 cells were incubated with various concentrations of edoxaban (0.1–100 µM), FSLLRY-NH2 (10 µM) as an inhibitor of PAR-2, or DPI (50 µM) as an inhibitor of NADPH oxidase for 30 min, followed by incubation with FXa (0.5 IU/mL), IS (500 µM), or AII (25 µM) for 3 h. Fluorescence intensity was measured at excitation and emission wavelengths of 485 and 535 nm, respectively, using a Tecan SPECTRA Fluor Plus microplate reader (Männedorf, Switzerland).

### 4.4. Electron Spin Resonance Spectroscopy

Direct radical scavenging activity for O_2_˙^−^ and •OH was evaluated by ESR spectroscopy. DMPO was used as a spin trap agent, and O_2_˙^−^ was generated by the xanthine–xanthine oxidase system. The final concentrations of reagents were DMPO (0.45 M), xanthine (100 µg/mL), and xanthine oxidase (0.02 U/mL). Exactly 1 min after mixing with or without various concentrations of edoxaban, ESR spectra were recorded at room temperature using a JEOL JES-FA100 ESR spectrometer (Tokyo, Japan) (power: 4 mW; center field: 335.0 mT; sweep width: 5 mT; modulate width: 0.7 mT; sweep time: 2 min; amplification: 300; time constant: 0.1 s). •OH was generated by UV photolysis of H_2_O_2_. The final concentrations of reagents were DMPO (9 mM) and H_2_O_2_ (0.5 mM). All solutions were prepared in a potassium phosphate buffer, pH 7.4. UV light was irradiated for 30 s after mixing with or without various concentrations of edoxaban, and ESR spectra were recorded at room temperature using a JEOL JES-X320 ESR spectrometer (power: 40 mW; Center field: 335.50 mT; sweep width: 5 mT; modulate width: 0.25 mT; sweep time: 2 min; amplification: 300; time constant: 0.3 s) after 30 s. After recording the EPR spectra, the signal intensities of DMPO-OH and the DMPO-OOH adducts were normalized against that of manganese oxide (Mn^2+^) signal, in which Mn^2+^ served as an internal control.

### 4.5. Measurement of Hydrogen Peroxide Decomposition

The concentration of hydrogen peroxide was measured as previously reported, with some modifications [29]. Various concentrations of edoxaban (1–100 µM) were added to a reaction mixture solution containing 100 µM H_2_O_2_, 120 mM KCl, and 50 mM Tris-HCl, pH 7.4. After incubating the samples at 37 °C for 10 min, the reaction was terminated by adding stopping solution (25 mg/mL of potassium biphthalate, 2.5 mg/mL NaOH, 82.5 mg/mL potassium iodide, and 0.25 mg/mL ammonium molybdate). The absorbance of the mixture was measured using a V-530 Jasco spectrophotometer (Japan Spectroscopic Co., Ltd., Tokyo, Japan) at 350 nm. The remaining hydrogen peroxide was detected using H_2_O_2_ solution as the standard.

### 4.6. Peroxynitrite Scavenging Assay Using Dihydrorhodamine123

Direct radical scavenging activity for ONOO^−^ was evaluated by monitoring the oxidation of DHR 123, as previously reported with some modifications [30,31]. ONOO^−^ was generated by decomposition of SIN-1. The final concentrations of reagents were 1 µM SIN-1 and 5 µM DHR123. The samples were incubated at 37 °C for 10 min after SIN-1 was mixed with or without different concentrations of edoxaban, and then DHR123 was added. ONOO^−^ scavenging by the oxidation of DHR 123 was measured using a Tecan Infinite 200 Pro microplate reader (Tecan, Maennedorf, Switzerland) at excitation and emission wavelengths of 485 and 535 nm, respectively, at room temperature. Because oxidation of DHR 123 was gradually increased by decomposition of SIN-1, fluorescence intensity was continuously measured for 10 min after adding DHR 123, and ONOO^−^ scavenging was calculated based on the average increase per minute.

### 4.7. Statistical Analysis

The results are reported as the mean ± S.D. Statistical analysis was performed using analysis of variance with the Tukey’s (Tukey–Kramer) test by Statcel3 (OMS publishing Inc., Saitama, Japan), an add-in software. For all analyses, *p* < 0.05 was regarded to indicate statistical significance.

## Figures and Tables

**Figure 1 ijms-20-04140-f001:**
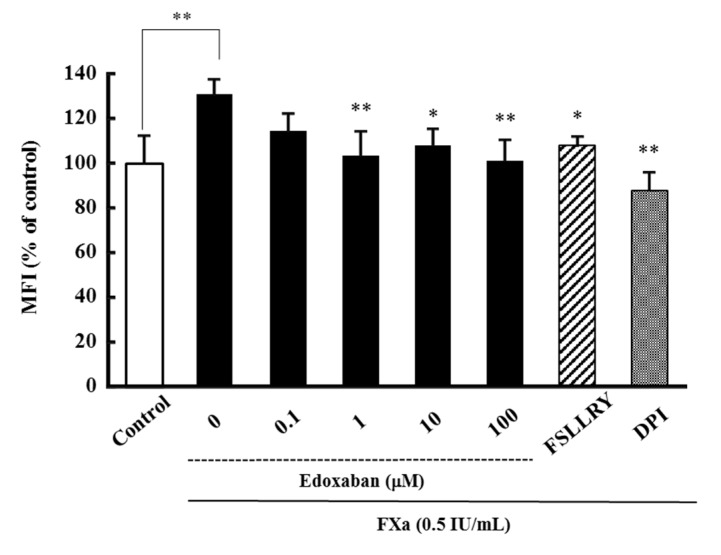
Effect of edoxaban-on FXa-derived intracellular reactive oxygen species (ROS) production in human proximal tubular cells (HK-2 cells). Effect of edoxaban (0.1–100 µM), Phe-Ser-Leu-Leu-Arg-Tyr-NH2 (FSLLRY-NH2) (10 µM) as an inhibitor of protease-activated receptor-2 (PAR-2), and diphenyleneiodonium chloride (DPI) (50 µM) as an inhibitor of nicotinamide adenine dinucleotide phosphate (NADPH) oxidase on FXa-induced intracellular ROS production in HK-2 cells. Values are expressed as the mean ± S.D. (*n* = 4). * *p* < 0.05, ** *p* < 0.01 vs. FXa alone.

**Figure 2 ijms-20-04140-f002:**
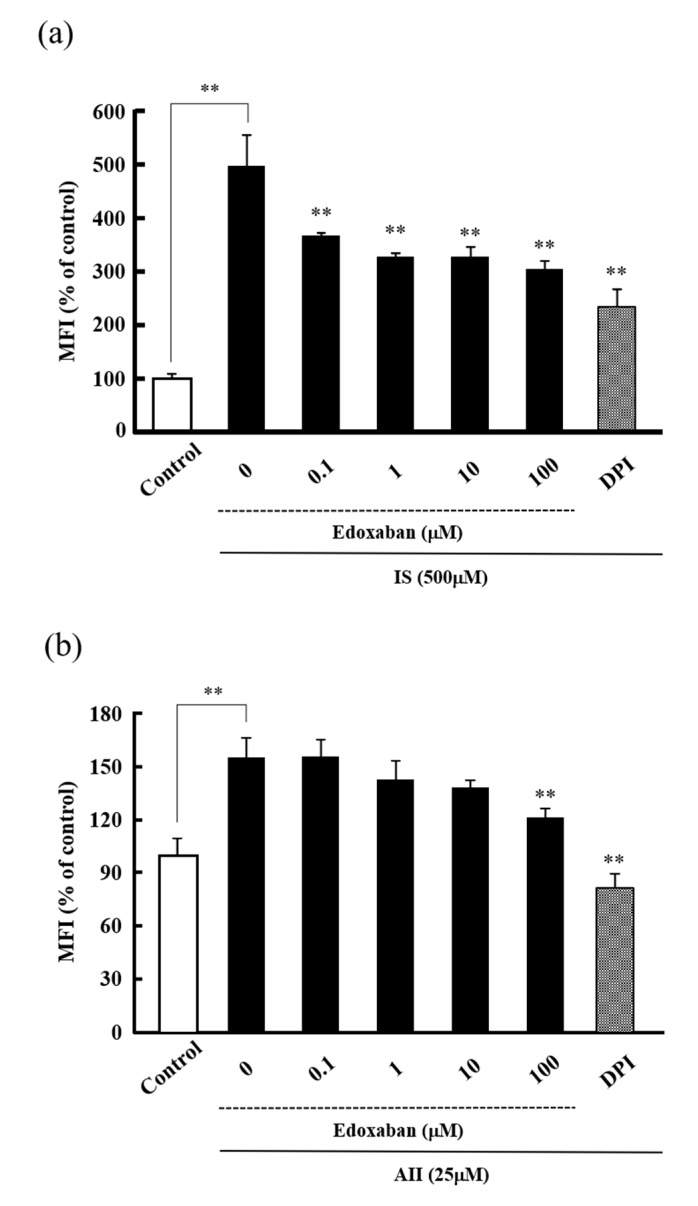
Effect of edoxaban on indoxyl sulfate (IS) or angiotensin II (AII)-derived ROS production in HK-2 cells. Effect of edoxaban (0.1–100 µM) and DPI (50 µM) as an inhibitor of NADPH oxidase on IS (**a**) or AII (**b**) derived intracellular ROS production in HK-2 cells. Values are expressed as the mean ± S.D. (*n* = 4). ** *p* < 0.01 vs. IS or AII alone.

**Figure 3 ijms-20-04140-f003:**
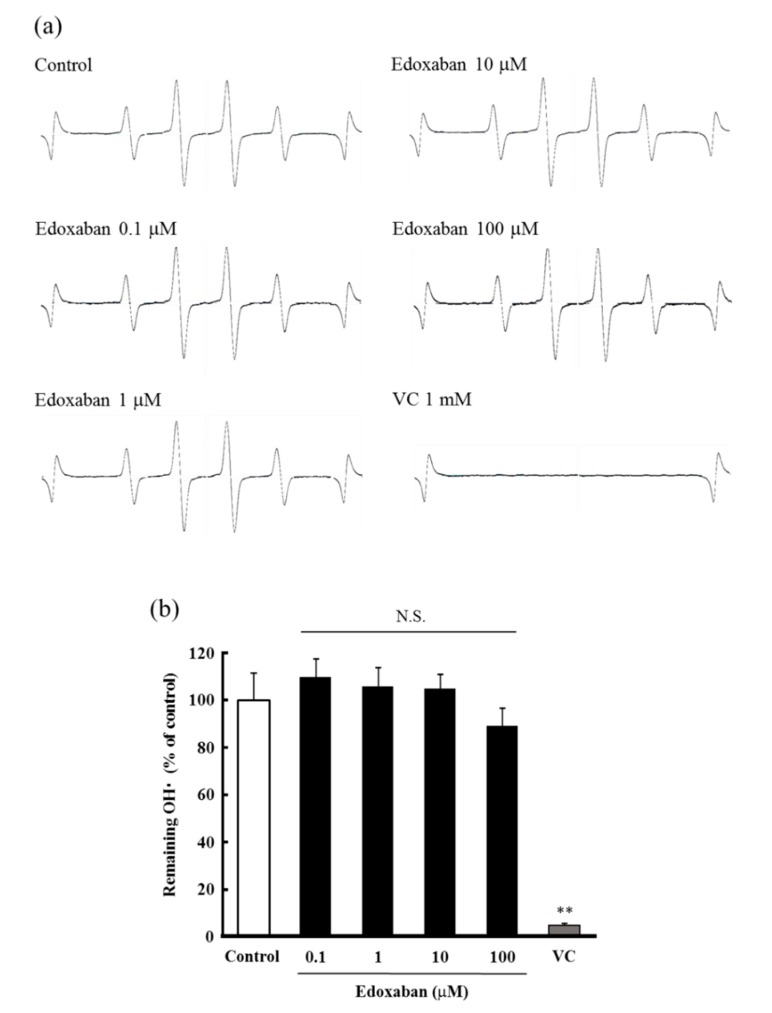
Effect of edoxaban on hydroxyl radical scavenging activity and hydrogen peroxide decomposition. (**a**) Electron spin resonance spectrum of 5,5-Dimethyl-1-pyrroline-*N*-oxide (DMPO)-OH signal in the reaction between DMPO and •OH. (**b**) Quantitation of •OH remaining. (**c**) Quantitation of H_2_O_2_ remaining. Values are expressed as the mean ± S.D. (*n* = 3). ** *p* < 0.01 vs. control. N.S.: not significant, VC: Vitamin C.

**Figure 4 ijms-20-04140-f004:**
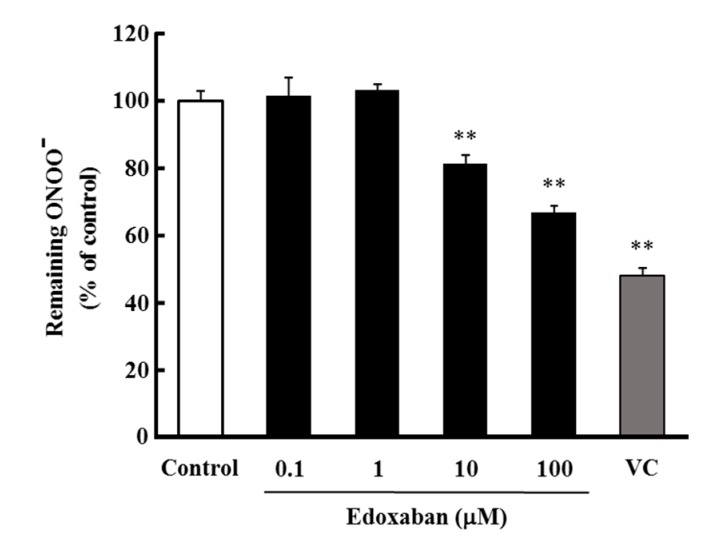
Effect of edoxaban on peroxynitrite (ONOO^−^) scavenging activity by dihydrorhodamine123 studies. Quantitation of ONOO^−^ remaining. Values are expressed as the mean ± S.D. (*n* = 4). ** *p* < 0.01 vs. control. VC: Vitamin C.

**Figure 5 ijms-20-04140-f005:**
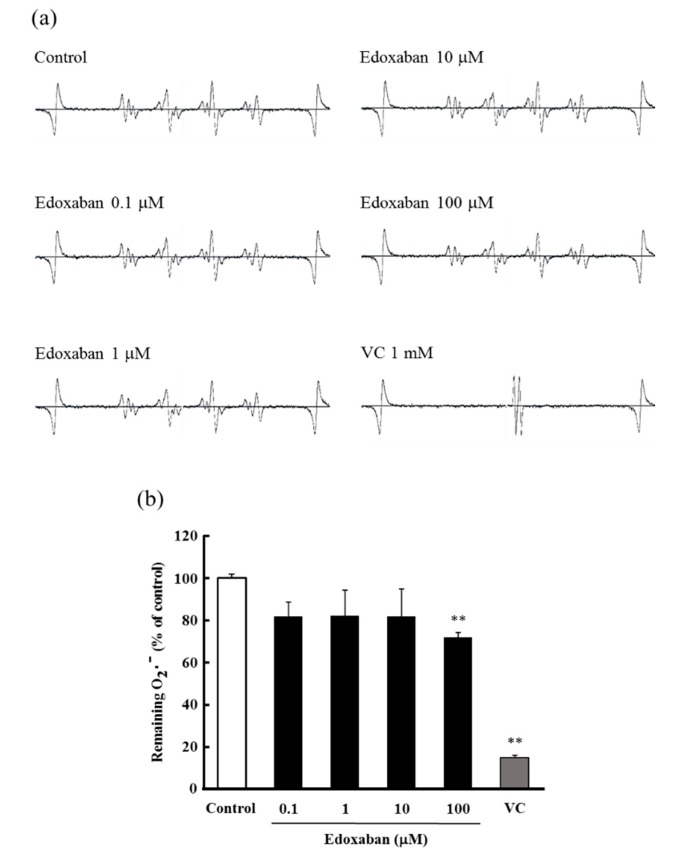
Effect of edoxaban on superoxide radical scavenging activity by electron spin resonance (ESR) spectroscopy. (**a**) ESR spectrum of DMPO-OOH signal in the reaction between DMPO and O_2_˙^−^. (**b**) Quantitation of O_2_˙^−^ remaining. Values are expressed as the mean ± S.D. (*n* = 3). ** *p* < 0.01 vs. control. VC: Vitamin C.

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
