# Peer review of "Edoxaban Exerts Antioxidant Effects Through FXa Inhibition and Direct Radical-Scavenging Activity"

_ijms, 2019, doi:10.3390/ijms20174140_

Round 1
Reviewer 1 Report
Authors have elucidated the role of edoxaban as a potential treatment for chronic kidney disease by eliciting antioxidant properties through inhibition of FXa and direct radical scavenging activity. Experimental design is good and technically manuscript is sound. Authors did a thorough job in measuring the ROS production through spectroscopy. However authors should provide more data to prove their point. I have some minor concerns which are as follows:
Please design some experiments regarding morphology of the cells with respect to ROS production. For superoxide and H2O2 show differential staining. For maintaining existing the tittle of the paper authors should provide more mechanistic details. More specifically some new experiments related to in vivo diabetic mice model, then only we can say that edoxaban is a potential molecule for treating CKD.Author Response
Please see the attachment.

Reviewer 2 Report
This in-vitro model on tubular cells demonstrates elegantly the capacity of Edobaxan to strongly mitigate oxydative stress at cellular level via both factor Xa pathway and other pleiotropic effects and its potential ability to lessen inflammation observed in CKD.
Authors should in their discussion add a paragraph on the adaptation of the dosage of the molecule in CKD patients not on dialysis (with a first step at moderate CKD and a second step when severe CKD is reached) together with its contraindication in those patients on dialysis (Jain N and Reilly F; Clinical pharmacology of oral anticoagulants in patients with kidney disease . CJASN 2019; 14: 278-287 )
